# Sexual health knowledge and needs among young adults with congenital heart disease

**Su Jin Kwon[1], Yu-Mi Im[2]\***

**1** Congenital Heart Disease Center, Asan Medical Center, Seoul, Korea, **2** Department of Nursing, Dankook University, Chungnam, Korea

\* ymim@dankook.ac.kr

## Abstract

Advances in the treatment of congenital heart disease (CHD) have increased life expectancy, entailing medical surveillance for a considerable number of adolescents and young adults with CHD for issues arising in areas such as sexual health. This study aimed to assess the sexual knowledge and the needs for sexual health education among this group. The participants comprised 53 young adult outpatients (27 males, median age: 23 years) who had undergone surgical interventions (median: 3 times) for CHD. The Knowledge related to Safe Sex Practice scale (KSSP), an assessment tool containing 15 questions on sexual knowledge, was administered, and the rates of correct answers for each item and the overall scale were compared with the age and sex of a control group ($n = 164$). The overall mean KSSP score of the participant group ($10.5 \pm 1.8$) was significantly lower than that of the control group ($11.1 \pm 1.9$, $p = .035$). The KSSP scores of the participants with low peripheral oxygen saturation ($SaO_2 < 95\%$) were significantly lower ($9.77 \pm 1.85$) than those with normal $SaO_2$ ($11.06 \pm 1.85$, $p = .009$). Regarding sexual health education, the participants reported receiving information about contraception as more important than other areas of sexual health. The rate of incorrect answers was higher for questions regarding natural ways of contraception utilizing infertile periods in the menstrual cycle. Based on an informed understanding of those with CHD, healthcare providers in this field should develop customized sexual health education for adolescents and young adults with CHD and implement customized sexual health education, including effective contraception methods.

## Introduction

Congenital heart disease (CHD) is the most common type among the major congenital anomalies, representing a major global health problem. Over time, the reported birth prevalence of CHD has increased substantially from less than one per 1,000 live births in the 1930s to nine per 1,000 live births in recent years [1]. Advances in medical and surgical management have increased the life expectancy of children with CHD to a more than 90% rate of survival into adulthood [2]. Therefore, offering optimal lifelong medical counseling to this high-risk population at various life stages has become important. Sexual and reproductive health is a key component for adolescents and young adults with CHD who require medical surveillance [3, 4].

at Asan Medical Center (irb@amc.seoul.kr; 82-2-3010-7166) or to the corresponding author.

**Funding:** This present research was supported by the research fund of Dankook University in 2019. This work was supported by the National Research Foundation of Korea (NRF) grant funded by the Korea government (MSIT) (No. 2018R1C1B5085730).

**Competing interests:** The authors have declared that no competing interests exist.

Sexual health is an important aspect of one's quality of life. The World Health Organization (WHO) defines sexual health as a "state of physical, emotional, mental, and social well-being in relation to sexual life," and identifies issues arising in relation to it as among the most important health issues that need to be resolved [5]. Despite its widely recognized importance, education to promote sexual health remains a sensitive and, occasionally, controversial issue. In Korea, public discussions about sex have traditionally been considered taboo, and, as part of a society in which the tenets of Confucianism retain considerable authority, Korean people tend to be reserved in expressing their sexual views and desires. However, with the rapid development of mass communication, Korean youth have gained easy access to cultures that are more liberal regarding sex, and, consequently, their attitudes towards sex are changing [6]. Nevertheless, there are limited data regarding the extent to which sexual and reproductive counseling is being provided to patients with CHD and the extent of their knowledge regarding the inherent risks of their specific cardiac conditions in relation to sexual and reproductive practices [7]. One study showed that only 54% of patients on an adult congenital cardiac clinic could identify their heart conditions correctly [8]. This number was found to be even lower among children and adolescents, with only 22% identifying their conditions accurately [9]. Although the reasons for this gap in their understanding are unclear, it is significantly different from that reported in similar populations with other chronic health issues. Due to this reduced awareness of their conditions, it is more likely that awareness of the risks associated with their specific heart conditions would also be limited among such patients. Additionally, awareness of safe sex, the possibility of pregnancy, pregnancy risk, and fetal toxicity of medications have been shown to be noticeably lacking [10].

There are limited studies in Korea related to sexual problems experienced by patients with chronic disease at various life stages, and few patients appear to acquire relevant knowledge regarding appropriate sexual behavior. Moreover, there are no accurate data regarding how such knowledge might be acquired, for example, whether through voice channels, the mass media, or the internet. However, appropriate sex counseling regarding contraception is important because unplanned pregnancies can have potentially fatal consequences for fetuses, mothers, or both, and, for those with CHD, recommendations for contraception should be made depending on an individual's heart condition [7, 11]. Healthcare providers in this field, who have previously built rapport and maintained good therapeutic relationships with patients, are likely to be the most appropriate health caregivers to act as educators and counselors to address such patients' sexual health problems. However, no study has been conducted on the sexual health of this patient group in Korea. Investigating the sexual health education experiences, needs, and knowledge of these patients will provide basic data for developing nursing interventions related to the current sexual health of these patients. Given our intention to develop an education program related to the sexual health of young adults with CHD, the purpose of this study was to identify their sexual health education experiences, sexual knowledge, and sexual health education needs.

## Materials and methods

### Study design and participants

This cross-sectional descriptive study enrolled 53 outpatients with CHD, aged between 18 and 37 years, who had attended a tertiary hospital located in Seoul and who were followed up by pediatric cardiologists or cardiac surgeons. For the purpose of comparison, this study included a control group comprising 164 healthy adults, who were of the same gender and ages as the patient group. They were recruited by flyers posted on social media sites such as Facebook.

## Questionnaires

**Characteristics of the participants.** The general characteristics of the participants with CHD were examined using six questions regarding age, gender, marital status, marital status, religion, and education level. Disease-related characteristics examined included CHD diagnosis and operation name (or procedure name) and frequencies, the presence or absence of drugs currently being taken, the number of drugs, oxygen saturation ($Sa0_2$) percentage, the New York Heart Association (NYHA) Functional Classification, and brain natriuretic peptide (BNP) levels, which were checked using medical records. Two sex-related questions assessed the attitudes of the participants' parents regarding their toleration of sexual behavior and the presence or absence of sexual experiences.

**Sexual health education experience.** Sexual health education provides knowledge and information related to sexual health including normal male and female reproductive systems, healthy and safe sex, various contraceptive measures, and sex-related diseases [5]. A questionnaire on sexual health education experiences, consisting of five questions regarding education experiences at schools, workplaces, and other remedial education programs, was administered. The questionnaire contained the following items: 1) "Have you ever received education regarding sexual health?"; 2) "How many times was the education provided?"; 3) "Please choose the content of the sexual health education you have received: 'Pregnancy and Childbirth,' 'Contraception,' 'Reproductive system and function,' 'Sexual harassment,' 'Gender identity,' 'Sexual counseling,' 'Sexual life,' 'Aging and sex,' 'Disability and sex,' 'Chronic diseases and sex,' or others" [12, 13]; 4) "Have you ever engaged in sexual intercourse?"; and 5) "Is there any information related to sex that you have been wanting to learn about?"

**Knowledge related to safe sex practice scale (KSSP).** The Korean version of the Knowledge related to Safe Sex Practice scale (KSSP), a tool for measuring sexual knowledge developed by Kong, Wu, and Loke [14], was used after obtaining approval from the developers and the translators [13]. At the time of the original author's tool development, the verified CVI (content validity index) was 0.93, and the Korean version TVI (translation validity index) measured by five nurses was 0.9 or higher. With respect to the reliability of the KSSP measuring tool, it was reported that there were no obvious inconsistencies in test-retest outcomes [13, 14]. This scale contains 15 items, and the questionnaire content concerns contraception and sexually transmitted diseases. Participants responded to the questions using the binary options of "right" or "wrong." One point was assigned for each question answered correctly, with higher scores indicating higher degrees of sexual knowledge.

## Data collection and ethical considerations

We received Asan Medical Center Institutional Review Board approval (IRB No. 2019–1504) and the research was conducted in accordance with the Declaration of Helsinki. We obtained permission and cooperation from the nursing department and outpatient clinic of the hospital's pediatric cardiology and cardiac surgery departments for conducting the survey. The contents of the questionnaire were explained face-to-face by the researchers before participants submitted written informed consent. We explained, in detail, the purpose and method of the study to participants and that they could withdraw their consent at any time during the study period. After providing them with sufficient time to decide whether to participate in the study, each participant submitted a written consent form. To maintain the privacy of the participants and allow them to answer sensitive questions freely, the survey was completed using online survey tools in a location of each participant's choosing. Healthy adults selected for comparison groups were identified using Google online questionnaires. Data collection was performed for a period of 11 months, from November 7, 2019, to September 11, 2020. The survey

questionnaire typically took 10 minutes to complete, and, on completion, participants received incentives in the form of mobile coffee coupons. The collected and related data were encrypted and stored on the researcher's password-protected computer.

## Statistical analysis

Data were analyzed using IBM SPSS Statistics 26.0 (IBM, South Korea). Descriptive statistics, including frequency, percentage, mean, standard deviation, Pearson's correlation, and an independent *t*-test, were used to compare the KSSP scores of the participant group with the control group; chi-squared tests were used to compare the correct answer rate for each question. Content analysis was performed on the descriptive responses to the question regarding the participants' sexual health education needs.

## Results

The sex and disease-related characteristics of the participants are shown in Table 1. A total of 53 outpatients with CHD, aged between 18 and 37 years, with an average age of 24.3 years, responded to the survey. Among the study participants, the percentage of each gender was approximately the same, 4 (7.5%) were married, and 49 (92.5%) were unmarried. Regarding their general characteristics, 32 (60.4%) had no religious affiliation and most were university students or graduates (67.9%). In terms of attitudes towards sex of their parents, 31 (58.5%) patients felt that they were allowed to have sex, and 25 (47.2%) had had sexual experiences. A total of 39 (73.6%) patients were taking up to six medications that mostly comprised anti-coagulants, vasodilators, and diuretics. In addition, almost all the study participants had undergone more than one operation and 12 (22.6%) participants had undergone more than four heart surgeries and procedures. In particular, 35 (66.0%) participants had functional single-ventricle physiology, which is a complex CHD that requires a minimum of three operations in newborn babies. Additionally, 41 (77.4%) participants were categorized as grade 1 (no symptoms and no limitation in ordinary physical activity), based on the NYHA Functional Classification. BNP, which can predict heart failure, showed normal levels in 29 participants (54.7%) whereas 31 (58.5%) showed $Sa0_2 < 95\%$. The participants had had 5.7 sexual education experiences on average and most had obtained their sexual health knowledge at schools or workplaces. The mean KSSP score associated with sexual health problems was 10.5 ± 1.8, with a maximum score distribution of 14 points and a minimum of 6. There were significant differences in participants' sexual knowledge scores by marital status ($p = .021$). $Sa0_2$, indicating the severity of the disease, was significantly correlated with KSSP scores ($p = .009$) (Table 1).

The overall KSSP mean score of the participant group was 10.5 ± 1.8, which was significantly lower than that of the control group (11.1 ± 1.5) ($t = -2.13$, $p = .035$). The control group had a mean age of 26.7 ± 4.6 years and 54% were men. Table 2 summarizes the number of correct responses obtained from the two groups, and shows the percentages of participants and healthy adults who answered individual questions correctly. Only four individuals in the control group gave correct responses to all 15 items. In addition, an item asking whether a condom could be reused after cleansing yielded a 100% correct response from the participant group. When comparing the correct answer rates for all 15 questions, a statistically significant difference was observed between the two groups for 6 questions. The results revealed statistically significant differences between groups in correct responses related to contraception methods, i.e., the appropriate method of removal of condoms after ejaculation (item 2, $\chi^2 = 5.03$, $p = .025$), the effectiveness of contraceptive pills (item 4, $\chi^2 = 6.20$, $p = .013$), contraceptive action time (item 10, $\chi^2 = 7.81$, $p = .005$), the correct use of vaginal douching (item 11, $\chi^2 = 5.12$, $p = .024$), the effectiveness of postcoital pills (item 12, $\chi^2 = 15.76$, $p < .001$), and the effectiveness of

**Table 1. Association of knowledge related to safe sex practice scores with participant characteristics.**

| Characteristics (N = 53) | | Mean ± SD or n (%) | KSSP[a] score (Mean ± SD) | t/F (p) or r (p) |
|---|---|---|---|---|
| **Age** | | 24.3 ± 4.5 | | r = .194 (.164) |
| **Sex** | Male | 27 (50.9) | 10.6 ± 1.8 | .111 (.912) |
| | Female | 26 (49.1) | 10.5 ± 1.9 | |
| **Married** | Yes | 4 (7.5) | 10.4 ± 1.8 | -2.37 (.021) |
| | No | 49 (92.5) | 12.5 ± 1.3 | |
| **Religion** | Christianity | 15 (28.3) | 11.1 ± 1.3 | 2.348 (.084) |
| | Buddhism | 3 (5.7) | 11. 7 ± 0.6 | |
| | Catholicism | 3 (5.7) | 8.7 ± 0.6 | |
| | None | 32 (60.4) | 10.3 ± 1.8 | |
| **Education** | High school graduate | 13 (24.5) | 10.9 ± 1.7 | 1.059 (.354) |
| | University student/graduate | 36 (67.9) | 10.3 ± 1.8 | |
| | Graduate school student | 4 (7.5) | 11.5 ± 2.4 | |
| **Family attitude towards sex** | Permissive | 31 (58.5) | 10.6 ± 2.0 | .402 (.689) |
| | Nonpermissive | 22 (41.5) | 10.4 ± 1.6 | |
| **Experience of coitus** | Yes | 25 (47.2) | 10.6 ± 2.0 | .271 (.788) |
| | No | 28 (52.8) | 10.5 ± 1.7 | |
| **Taking medication at present** | Yes (up to 6 types) | 39 (73.6) | 10.4 ± 1.8 | -.619 (.539) |
| | No | 14 (26.4) | 10.8 ± 1.9 | |
| **Number of surgeries and interventions** | One | 4 (7.5) | 11.5 ± 2.4 | 1.474 (.233) |
| | Two | 19 (35.8) | 10.6 ± 2.0 | |
| | Three | 18 (34.0) | 9.9 ± 1.8 | |
| | More than four | 12 (22.6) | 11.0 ± 1.0 | |
| **NYHA[b] Functional Classification** | 1 | 41 (77.4) | 10.4 ± 1.9 | -.846 (.402) |
| | 2 | 12 (22.6) | 10.9 ± 1.6 | |
| **Diagnosis** | Functionally single-ventricular physiology | 35 (66.0) | 10.5 ± 1.9 | .081 (.939) |
| | Biventricular physiology | 18 (34.0) | 10.5 ± 1.8 | |
| **BNP[c] levels** | Normal (< 29 pg/ml) | 29 (54.7) | 10.4 ± 2.1 | .467 (.643) |
| | Abnormal (≥ 29 pq/ml) | 24 (45.3) | 10.7 ± 1.3 | |
| **Sa0$_2$[d] (%)** | Normal (≥95) | 22 (41.5) | 11.1 ± 1.6 | .466 (.009) |
| | Desaturation (<95) | 31 (58.5) | 9.8 ± 1.9 | |
| **Number of sexual health education** | | 5.7 ± 3.4 | | r = .033 (.815) |

[a] KSSP: Knowledge Related to Safe Sex Practice

[b] NYHA: New York Heart Association

[c] BNP: brain natriuretic peptide

[d] oxygen saturation.

engaging in sexual activities during the safe period (item 13, $\chi^2 = 7.64$, $p = .006$). In particular, the rate of incorrect answers was higher for KSSP questions related to natural ways of contraception such as utilizing infertile periods in the menstrual cycle.

The participant group's prior experiences of sexual health education are summarized in Table 3. All areas of sexual health education content were included, resulting in multiple responses for the participants. Forty-two (79.2%) participants had received education on the structure and function of the reproductive system, 40 (75.5%) on pregnancy and childbirth, 37 (69.8%) on sexual harassment, and 33 (62.3%) on contraception. Education related to disabilities and sex and chronic diseases and sex had rarely been provided. A total of 35.8% of the

**Table 2. Knowledge related to safe sex practices in participants and healthy adults.**

| Safe sex practice knowledge items (T = true, F = false) | Number (percentage) of correct answers | | | |
|---|---|---|---|---|
| | Participants (N = 53) n (%) | Controls (N = 164) n (%) | Overall (N = 217) n (%) | $\chi^2$ |
| 1. When using a condom, make sure to leave some space near its tip (T) | 40 (75.5) | 118 (72.0) | 158 (72.8) | 0.25 |
| | | | | $p = .617$ |
| 2. Sexual partners need to remove condoms by rolling them from their bases, post ejaculation, to prevent semen from leaking out (T) | 32 (60.4) | 70 (42.7) | 102 (47.0) | 5.03 |
| | | | | $p = .025^*$ |
| 3. The reuse of a condom is possible after a thorough washing (F) | 53 (100.0) | 162 (98.8) | 215 (99.1) | 0.65 |
| | | | | $p = 0.419$ |
| 4. Daily prevention pills are an effective method of contraception (T) | 23 (43.4) | 103 (62.8) | 126 (58.1) | 6.20 |
| | | | | $p = .013^*$ |
| 5. Two days is the lifespan of the male sperm inside a woman's body (F) | 42 (79.2) | 126 (76.8) | 168 (77.4) | 0.13 |
| | | | | $p = 0.715$ |
| 6. Pregnancy is an unlikely result of adolescents taking part in sexual intercourse for the first time (F) | 44 (83.0) | 150 (91.5) | 194 (89.4) | 3.01 |
| | | | | $p = .083$ |
| 7. Taking the penis out of the vagina right before ejaculation is a safe and reliable way of preventing pregnancy (F) | 48 (90.6) | 156 (95.1) | 204 (94.0) | 1.48 |
| | | | | $p = .224$ |
| 8. Having sexual intercourse with an individual under the age of 16 is against the law (T) | 37 (69.8) | 101 (61.6) | 138 (63.6) | 1.17 |
| | | | | $p = .279$ |
| 9. The withdrawal method removing the penis just before ejaculation is the best way of preventing STDs (sexually transmitted diseases) (F) | 47 (88.7) | 154 (93.9) | 201 (92.6) | 1.60 |
| | | | | $p = .206$ |
| 10. Once a person has taken their first contraceptive pill, they are immediately safeguarded from getting pregnant (F) | 43 (81.1) | 154 (93.9) | 197 (90.8) | 7.81 |
| | | | | $p = .005^*$ |
| 11. Pregnancy can be avoided by regular vaginal douching (F) | 42 (79.2) | 149 (90.9) | 191 (88.0) | 5.12 |
| | | | | $p = .024^*$ |
| 12. The consumption of a "morning after pill" within seven days of intercourse can prevent pregnancy (F) | 37 (69.8) | 150 (91.5) | 187 (86.2) | 15.76 |
| | | | | $p \leq .001^{***}$ |
| 13. Avoiding sexual intercourse according to the menstrual cycle is an effective contraception method (F) | 28 (52.8) | 120 (73.2) | 148 (68.2) | 7.64 |
| | | | | $p = .006^*$ |
| 14. The safe period for not getting pregnant is seven (7) days after and seven (7) days before menstruation (T) | 25 (47.2) | 56 (34.1) | 81 (37.3) | 2.90 |
| | | | | $p = .088$ |
| 15. Sexual intercourse during menstruation does not result in pregnancy (T) | 17 (32.1) | 46 (28.0) | 63 (29.0) | 0.32 |
| | | | | $p = .574$ |

\* $p < .05$

\*\* $p < .01$

\*\*\* $p < .001$.

For total mean scores, $t$-test = -2.13, $p = .035$.

respondents believed that contraception was the most important part of their education content.

## Discussion

This study was conducted to identify the sexual health education experiences, knowledge, and needs of selected young adults with CHD. Most of them had complex heart conditions (for example, functional single-ventricle physiology or tetralogy of Fallot) and had undergone hospital-based surgeries and treatments since birth. Since it is difficult to openly discuss sex in

**Table 3. Sexual health education contents for participants with congenital heart disease (CHD).**

| Contents (N = 53) | Sexual health education Responses[a] N (%) | Perceived importance N (%) |
|---|---|---|
| Pregnancy and childbirth | 40 (75.5) | 15 (28.3) |
| Contraception | 33 (62.3) | 19 (35.8) |
| Reproductive system and function | 42 (79.2) | 8 (15.1) |
| Sexual harassment | 37 (69.8) | 14 (26.4) |
| Gender identity | 24 (45.3) | 6 (11.3) |
| Sexual counseling | 12 (22.6) | 2 (3.8) |
| Sexual life | 25 (47.2) | 6 (11.3) |
| Aging and sex | 0 (0) | 0 (0) |
| Disability and sex | 2 (3.8) | 1 (1.9) |
| Chronic disease and sex | 3 (5.7) | 0 (0) |
| Miscellaneous[b] | 0 (0) | 4 (7.5) |

[a] Multiple response data.

[b]Miscellaneous educational content: "the influences of congenital heart disease on sex," "whether congenital heart disease can be inherited from patients," "optimal contraceptive measures for patients with congenital heart disease," "sex-related diseases," and "marital life."

Korean society, and there is a lack of research examining how individuals with CHD can maintain their sexual health in relation to their conditions, this study provides basic data for the development of educational materials to provide more targeted sexual health care.

According to a recent study conducted in Korea, the total percentage of adolescents who were found to be sexually active ranged from 5.0% to 5.3% [15]. The study also noted that the mean age for Koreans' first sexual experience was between the ages of 12.8 and 13.2, which remained constant over the three-year period of the study. Lastly, among sexually active Korean female participants, 0.2% were reported to have experienced pregnancy, most of whom (66.1% to 73.6%) had undergone an induced abortion [15]. Hence, sexual health education is very important in helping to raise the age at which sexual behavior begins. Whether teenagers with chronic diseases, especially those with CHD, have appropriate sex-related knowledge and have received sexual health education is likely to affect when they begin to have sex. However, most young adults with CHD who participated in this study reported not having received any sexual health education related to their diseases.

The mean KSSP score of the participants was 10.5 ± 1.8, which was significantly lower than that of the control group (11.1 ± 1.5). The higher the CHD severity, the lower the likelihood of participating in normal school life and education due to frequent surgeries and hospitalizations. In addition, the more severe the symptoms caused by heart disease, the lower a person's interest in sex. Thus, it was anticipated that the level of knowledge regarding sex would be lower in the participant group. A specific sexual knowledge questionnaire for patients with CHD was not available, so the difference between the participant and control groups had to be evaluated using the same questionnaire as used for people generally. When the sexual knowledge of clinical nurses [13] and nursing students [14] was measured using the same tool, the average score was 11 and 12.4, respectively. The participants in this study had a lower average sexual knowledge score compared to nurses, which could be explained as due to the latter having more opportunities for receiving sex-related education given the nature of their job. However, it could also be the case that the participants with CHD, who had undergone several surgeries from a young age, would have decreased interest in and knowledge of sex compared to healthy adults. In this study, 47.2% of the young adults reported having had sexual

experiences, which is in line with the results of previous studies indicating that patients with CHD have a low sexual intercourse experience rate in comparison to healthy individuals of a similar age [16, 17]. CHD may result in parents raising their children with special care. In line with our findings, previous studies have found that only 47.2% of patients with CHD reported engaging in sexual activities, which is less than that reported in the general healthy adult population (82.4%); this difference might be due to parents' protective attitudes or because patients with CHD feel self-conscious about their body image due to their physical scars from surgeries [16, 18]. In contrast, another study found that there are few differences in the sexual behavior of individuals with or without CHD. The age of initial intercourse, the number of sexual partners at various time intervals, and the number of sexual encounters without condoms did not differ between the two groups [7].

Although it is generally understood to be difficult to initiate sexual relations in Korean culture, additional research is needed regarding sexuality. All individuals have potential sexuality and sexual health therapy needs. However, these needs are likely to differ somewhat between those with and without CHD, as being sexually active entails a greater health risk among those with CHD. Despite the clear need for intervention, there is currently a lack of development and employment of sexual and reproductive health counseling for those with CHD [7]. Furthermore, sexual health nursing is a complex area with multiple facets and is particularly difficult to work in effectively because of barriers arising within traditional Confucian cultures such as Korea.

The KSSP scores of the participants with low peripheral $SaO_2$ ($Sa0_2 < 95\%$) were significantly lower (9.8 ± 1.9) than those with normal $SaO_2$ (11.1 ± 1.6, $p$ = .009). Low $SaO_2$ levels were found in participants with severe CHD who could only engage in limited activities. Therefore, it appears that when CHD was more severe, the sexual interest of those individuals was decreased and lower KSSP scores obtained.

In this study, most young adults with CHD had received sexual health education at school, work, and elsewhere, at an average of 5.7 times; however, they had not received customized sexual health education regarding contraceptive methods, pregnancy, and childbirth that was appropriate for their diseases. The correct answer rate for contraceptive methods was significantly lower among the participants in this study compared to the healthy adults. Regarding sexual health education, the participants reported learning about the reproductive system most frequently; however, they perceived information about contraception as being more important. Nevertheless, the rate of incorrect answers was higher for the KSSP questions regarding natural methods of contraception such as utilizing infertile periods in the menstrual cycle. Appropriate sex counseling regarding contraception is generally important given the multiple challenges of pregnancy but also needs to consider an individual's heart condition where relevant [7, 11]. In particular, the high estrogen levels in oral contraceptives increase the risk of thrombosis, and such contraceptives are not recommended for patients with Eisenmenger syndrome and ventricular dysfunction or for those who have undergone valve replacement or a Fontan procedure [19]. Only 21% of adolescents with CHD have been found to have appropriate knowledge of contraception methods that were most suited for their conditions, with only 25% of them identifying health care providers as the source of this knowledge [20, 21]. Lack of appropriate counseling could potentially have significantly negative health effects in relation to pregnancy, particularly for those who have undergone a Fontan procedure [7].

As the average age of individuals at the time of their first sexual activity in this study was reported to be as early as 12 years, sexual health education should be provided earlier than the age of 12. The contents of an educational program should be appropriate for each age group, since knowledge, comprehension, and attitudes towards sexual activity may vary with different ages [15]. Considering the high incorrect response rate for contraception methods in this

study, it is especially important to start sexual health education (including education concerning birth control methods suitable for one's illness) before adolescence, when sexual initiation tends to occur. If adolescents start engaging in sex without knowing appropriate contraceptive methods, unprotected sexual activity could lead to unwanted teenage pregnancies, which are associated with higher maternal and neonatal morbidity and mortality cases, an increased likelihood of unsafe abortions, and poorer lifelong health and fertility outcomes. In addition, becoming pregnant while still an adolescent may affect one's social and educational life due to social pressures that can be imposed on such adolescents and because they may be viewed as bad influences on other adolescents in their community [15].

In terms of sexual health education content, the participants had experienced education most often on the structure and function of reproductive organs (79.2%) followed by pregnancy and childbirth (75.5%), whereas education on chronic diseases and sex (5.6%) and disability and sex (3.7%) had been much less often experienced. Pregnancy and childbirth, the structure and function of reproductive organs, and contraception are the basics of sexual health education and most participants had been educated mainly on the physical aspects of sex. However, despite having chronic diseases, very few of the participants had received sexual health education related to their conditions. Furthermore, previous studies have contended that it is preferable for the contents of sexual health education to encompass social and mental aspects such as sexual diversity, life-cycle sex problems, sexual orientation, and gender identity, and to address emotional aspects such as sexual pleasure and physical intimacy, as well as behavioral aspects such as sexual behavior, response, and function [22, 23]. Although targeted sexual health education is needed in varying contexts, no education program is available for these types of patients in Korea; therefore, research on and development of relevant sexual health education programs should be conducted.

In addition, if the needs of those targeted for sexual health education are evaluated and reflected on prior to providing sexual health education programs, the content is likely to be more effective. For example, unlike the situation for healthy adults, those with CHD may well have concerns regarding whether one's heart could be adversely affected by sexual excitement and whether one's child might be negatively affected during pregnancy. It has been reported that decreased sexual behavior or sexual dysfunction may occur in patients with heart failure; thus, sexual health education tailored for individual heart conditions is necessary. Moreover, it is important to create an environment where one can ask medical staff to address individual concerns without the individuals involved feeling ashamed. In addition, for women with CHD, pregnancy can cause hemodynamic changes and complications in their hearts and fetuses; hence, close assessment and education should be provided in conjunction with other departments such as obstetrics and gynecology [11, 24].

Based on the results of this study, young adults with CHD need educators and counselors to address their sexual health problems with appropriate sexual knowledge and responsible attitudes, especially as the current sexual health education content lacks adequate information concerning contraception despite high educational needs in this area. Therefore, healthcare providers in this field should have detailed sexual knowledge, the communication skills necessary for sexual health assessment, and the ability to perform sexual health nursing, while comprehensively dealing with sexual behavior, satisfaction, and feelings, which would help address deficiencies in current sexual health education.

## Conclusion

This study is the first in Korea to identify the sexual health education experiences, knowledge, and needs of young adults with CHD. The results of this research suggest that customized

sexual health education should be developed for adolescents and young adults with CHD. Based on an informed understanding of these individuals' diseases, healthcare providers in this field should implement customized sexual health education, including providing information on appropriate contraception methods. Furthermore, it is important that counseling and education sessions are conducted for adolescents as it is during adolescence that sexual intercourse is often initiated, and that sexual health education begin in schools prior to adolescence. However, one limitation of the study was that, since the participants were outpatients with complex CHD who were recruited from a hospital using convenience sampling, the sample used in this study may not be representative of the overall population with CHD.

## Supporting information

**S1 Questionnaire.**
(DOCX)

## Author Contributions

**Conceptualization:** Yu-Mi Im.

**Data curation:** Yu-Mi Im.

**Investigation:** Su Jin Kwon.

**Methodology:** Yu-Mi Im.

**Project administration:** Yu-Mi Im.

**Supervision:** Yu-Mi Im.

**Writing – original draft:** Su Jin Kwon.

**Writing – review & editing:** Yu-Mi Im.

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
