## [Editor Report · Decision Letter 0]

9 Feb 2021

PONE-D-21-03020

Sexual Health Knowledge and Needs Among Young Adults with Congenital Heart Disease

PLOS ONE

Dear Dr. Im,

Thank you for submitting your manuscript to PLOS ONE. After careful consideration, we feel that it has merit but does not fully meet PLOS ONE’s publication criteria as it currently stands. Therefore, we invite you to submit a revised version of the manuscript that addresses the points raised during the review process.

We look forward to receiving your revised manuscript.

Kind regards,

Ka Ming Chow

Academic Editor

PLOS ONE

Journal Requirements:

3.Thank you for including your ethics statement: "We received IRB approval (IRB No. 2019-1504) of 'A' hospital in seoul and obtained permission from all participants. "

4.We note that you have indicated that data from this study are available upon request. PLOS only allows data to be available upon request if there are legal or ethical restrictions on sharing data publicly. For information on unacceptable data access restrictions, please see http://journals.plos.org/plosone/s/data-availability#loc-unacceptable-data-access-restrictions.

5. Please provide additional details regarding participant consent. In the ethics statement in the Methods and online submission information, please ensure that you have specified what type you obtained (for instance, written or verbal, and if verbal, how it was documented and witnessed). If your study included minors, state whether you obtained consent from parents or guardians. If the need for consent was waived by the ethics committee, please include this information.

Additional Editor Comments:

Thank you for inviting me for reviewing this paper. The objective of the paper was to assess the sexual knowledge and the need for sexual health education of young adults with congenital heart disease in Seoul. The study filled the gap in which sexual health and awareness of such need are limited. My comments on the paper are summarised as follows:

1. The participants in the intervention group and control group should receive similar knowledge on sexual health from school, why there is difference in sexual knowledge between the groups? The same generic sexual health questionnaire was adopted to measure knowledge in both groups. Why there is significant difference in the result? It would be better to explain such phenomenon clearly in the discussion part.

2. Is there any significant difference in demographic data between the intervention and control groups, e.g. age, family attitude towards sex, sexual experience, gender and marital status? Is the significant result in sexual knowledge between groups due to the difference in demographic data between groups? It would be clearer to have such data analysis and findings reporting.

3. With revision on the above comments, the findings of the study will provide implications for the design of sexual health education for this population of patients.
---

## [Author Response · Author response to Decision Letter 0]

30 Mar 2021

Dear Reviewer, 

Thank you for taking the time to review our manuscript. We have revised the manuscript according to your feedback and hope it is satisfactory.

Should you have any questions or comments, please do not hesitate to let us know.

Sincerely,

The authors

---

## [Editor Report · Decision Letter 1]

1 Apr 2021

PONE-D-21-03020R1

Sexual Health Knowledge and Needs Among Young Adults with Congenital Heart Disease

PLOS ONE

Dear Dr. Im,

Thank you for submitting your manuscript to PLOS ONE. After careful consideration, we feel that it has merit but does not fully meet PLOS ONE’s publication criteria as it currently stands. Therefore, we invite you to submit a revised version of the manuscript that addresses the points raised during the review process.

ACADEMIC EDITOR:

Thank you for revising the manuscript according to reviewers' comments. The revision of the manuscript has addressed most of our enquires. However, there are still comments or suggestions for the revision of the manuscript, which are listed as follows:

1. Although it is mentioned in the discussion part that unplanned pregnancies can have potentially fatal consequences for fetuses, mothers, or both, risk of pregnancy and induced abortion in people with CHD should be explicitly stated in the introduction to justify the significance of adequate sexual education for this group of people and the need to conduct this study to identify their sexual knowledge.

2. Suggest to add reliability and validity of KSSP on page 6.

3. Suggest to add "indication of grade 1 based on the NYHA Functional Classification" in line 162 on page 8.

4. Use the abbreviation "CHD" in line 236 on page 15 to replace congenital heart disease.

We look forward to receiving your revised manuscript.

Kind regards,

Ka Ming Chow

Academic Editor

PLOS ONE
---

## [Author Response · Author response to Decision Letter 1]

17 Apr 2021

April 14, 2021

Dear Reviewer, 

Thank you for taking the time to review our manuscript. We have revised the manuscript according to your feedback and hope it is satisfactory.

Should you have any questions or comments, please do not hesitate to let us know.

Yours sincerely,

The authors

---

## [Editor Report · Decision Letter 2]

21 Apr 2021

Sexual Health Knowledge and Needs Among Young Adults with Congenital Heart Disease

PONE-D-21-03020R2

Dear Dr. Im,

We’re pleased to inform you that your manuscript has been judged scientifically suitable for publication and will be formally accepted for publication once it meets all outstanding technical requirements.

Kind regards,

Ka Ming Chow

Academic Editor

PLOS ONE

---

## [Editor Report · Acceptance letter]

26 Apr 2021

PONE-D-21-03020R2 

Sexual Health Knowledge and Needs Among Young Adults with Congenital Heart Disease 

Dear Dr. Im:

I'm pleased to inform you that your manuscript has been deemed suitable for publication in PLOS ONE. Congratulations! Your manuscript is now with our production department. 

Kind regards, 

on behalf of

Dr. Ka Ming Chow 

Academic Editor

PLOS ONE